# Change in Depression and Its Determinants during the COVID-19 Pandemic: A Longitudinal Examination among Racially/Ethnically Diverse US Adults

**DOI:** 10.3390/ijerph19031194

**Published:** 2022-01-21

**Authors:** Yuzi Zhang, Kathryn M. Janda, Nalini Ranjit, Deborah Salvo, Aida Nielsen, Alexandra van den Berg

**Affiliations:** 1UTHealth School of Public Health, Austin, TX 78701, USA; kathryn.m.janda@uth.tmc.edu (K.M.J.); nalini.ranjit@uth.tmc.edu (N.R.); aida.nielsen@uth.tmc.edu (A.N.); alexandra.e.vandenberg@uth.tmc.edu (A.v.d.B.); 2Michael and Susan Dell Center for Healthy Living, Austin, TX 78701, USA; 3Prevention Research Center, Brown School, Washington University in Saint Louis, Saint Louis, MO 63130, USA; deborah.salvodominguez@uth.tmc.edu

**Keywords:** depression, racial/ethnic minorities, health behaviors, longitudinal design, COVID-19, mental health, physical activity, sedentary time

## Abstract

This study examined longitudinal data to identify changes in the occurrence of depressive symptoms, and to explore if such changes were associated with socio-demographic, movement behaviors, and health variables during the COVID-19 pandemic, among a diverse sample of central Texas residents. Participants who completed two online surveys in 2020 (in June and November) from an on-going longitudinal study were included. Depressive symptoms were measured by Patient Health Questionnaire-2. Change in depressive symptoms’ occurrence status between the two time points was categorized into (1) stable/improved, and (2) consistent depressive symptoms/declined. Sociodemographic factors, movement behaviors and health data were self-reported. Statistical analyses utilized descriptive statistics and logistical regression. Among a total of 290 individuals (84.1% female; 71.0% racial/ethnic minorities), 13.5% were categorized as consistent depressive symptoms/declined. Multivariable logistic regression indicated that racial/ethnic minorities, older age, and increased physical activity were associated with a lower likelihood, while greater sedentary time was associated with higher likelihood of consistent depressive symptoms/declined status. Between 3 months and 8 months into the pandemic, various socio-demographic and behavioral variables were associated with changes in depressive symptoms’ occurrence status. Future research should explore the longer-term impacts of COVID-19 on depression among a diverse population and identify risk factors for depression.

## 1. Introduction

### 1.1. COVID-19 and Mental Health Consequences

The global Coronavirus Disease 2019 (COVID-19) pandemic changed multiple aspects of daily living. Aside from being worried about getting and spreading the disease, individuals have been facing new challenges, such as the uncertainty of employment and income, restrictions on social interaction, and for many working and studying online constantly. The pandemic has affected both physical health and mental health. A systematic review which combined evidence from different countries has shown that a high prevalence of adverse mental health symptoms was reported during COVID-19 among the general population [1]. Among US adults, the prevalence of depression in each severity category increased during the early stages of the pandemic (March–April 2020), compared with pre-pandemic times [2]. It is possible that the prevalence of depression during the early months of COVID-19 in the US may have continued to increase as the pandemic continued. A repeated cross-sectional study by Coley et al. found that the prevalence of depression among US adults increased from 32% in late April/early May 2020 to 44% in mid-November 2020 [3].

### 1.2. Variables Related to Depression during the Pandemic

Similar to other health conditions, the mental health of racial/ethnic minorities may have been disproportionately affected by COVID-19. McKnight-Eily et al. found that the percentage of stress related to food insecurity and stable housing was higher among Hispanic versus White adults (22.7% vs. 11.9%, and 20.7% vs. 9.2%, respectively) [4]. The Hispanic respondents reported higher levels of COVID-19-related fear, which was directly associated with both depression and anxiety symptoms [5]. A cross-sectional study by Saltzman et al. conducted from April to May 2020 found that Hispanic respondents were 10 times more likely to be classified as having depression compared to non-Hispanic Whites [6].

Other variables, such as age, occupation, and financial issues also had impact on depression status during the pandemic. A cross-sectional study during 24–30 June 2020, assessed the mental health of US adults and found that the prevalence of depression was 52.3%, 32.5%, 14.4%, and 5.8%, for respondents aged 18–24, 25–44, 45–54, and 65 and above years, respectively [7]. Individuals with financial or food access concerns [8], lower household income [9], lower household savings [2], and those who no longer worked due to COVID-19 [2,8], were more likely to experience symptoms of depression. In addition, those living with children at home had 1.42 times the odds of depression compared to people without children in the household [9].

The implementation of preventive measures, such as social distancing, shelter-in-place, and work-from-home, have limited daily activities and in-person interactions. A recent review focusing on physical activity and sedentary behavior during the COVID-19 pandemic among employed adults reported consistent evidence of decreased overall physical activity levels and increased total sedentary behaviors across most of the studies from various countries [10]. Benefits of physical activity in preventing or treating depression have been shown in both observational and experimental studies [11]. A recent systematic review during the COVID-19 pandemic indicated that individuals who spent more time in moderate- to vigorous-intensity physical activity were less likely to have depression, and decreased physical activity was associated with greater depressive symptoms [12]. Sedentary behavior, such as screen time, on the other hand, is known to be associated with greater risk of depression [13].

### 1.3. Gaps in the Literature

To date, few studies have examined data from repeated cross-sectional studies [3] or compared the current cross-sectional data with previously collected data [2] among the US population to examine trends over time. The aforementioned studies and other cross-sectional studies provided timely measures of prevalence that have reflected the burden of depression and raised awareness of the importance of mental health. However, such studies are subject to a substantial likelihood of confounding bias from sample composition differences, and as such may provide inaccurate estimates of changes in depression over time. Therefore, longitudinal research using prospective data is necessary in order to understand changes in depression during the COVID-19 pandemic.

Most of the longitudinal research concerning mental health during the COVID-19 pandemic has been conducted outside the US, with evidence published from the United Kingdom [14,15], Germany [16,17], and Canada [18]; moreover, most of this data was collected between March and June 2020 and focused on immediate impacts of the onset of the pandemic. One longitudinal study conducted among US residents measured mental health outcomes, and various socio-demographic as well as health behavioral predictors at three time points (about two weeks apart for each measure) between April and May 2020 [19]. Their findings suggest that, during the early months of the pandemic, the prevalence of being depressed or depression severity levels were highest in the first wave and gradually became less severe over the remaining two waves [19]. Nevertheless, the medium-term effects of COVID-19 on depression among the US general population remain unclear. In the aforementioned longitudinal study, the sample was predominantly Whites/Caucasians, a group that as a whole experienced fewer effects. Minority race/ethnicity being underrepresented is not uncommon in the previous literature, regardless of study design. Therefore, gaining insights into the changes in depression status during the later months of the COVID-19 pandemic in a socioeconomically, racially and ethnically diverse sample, and exploring which factors are related to these changes, can provide valuable information for designing effective interventions to improve the mental health of the most vulnerable US populations.

### 1.4. Purpose of the Study

To address previous research gaps, the current study used longitudinal data, between the first 3 months and 8 months of the onset of the COVID-19 pandemic, to identify patterns of change in the occurrence of depressive symptoms. We further explored the association between socio-demographic, behavioral, and health outcome variables, and depressive symptom occurrence patterns during the pandemic, among a diverse sample of central Texas residents.

## 2. Materials and Methods

### 2.1. Study Sample

The Food Retail: Evaluating Strategies for a Healthy Austin (FRESH Austin) project is a 3-year longitudinal study evaluating the impact of several coordinated food access strategies in underserved communities in Austin on healthy food purchasing and consumption [20]. Data were collected via a yearly survey from a cohort of 400 low-income residents of Austin, Texas. During the last year of the study, to better understand the impact of Coronavirus Disease 2019 (COVID-19) on food-related behaviors, two online surveys (in June and November 2020) were distributed to the original FRESH Austin cohort via email or text message, and all survey and recruitment messages were provided in English and Spanish. All participants who completed the first survey had an opportunity to refer a friend or a family member outside their household to take a survey. A $20 gift card was given to participants for the completion of each survey. In addition, a $5 gift card was emailed to all who referred a potential participant, once that referred person had completed a survey. A total of 367 and 304 individuals completed the first and the second survey, respectively. The current study includes 290 participants who completed both surveys and supplied complete data on the Patient Health Questionnaire-2 questions on both surveys. Participants provided consent to participate in each survey. All procedures were approved by the University of Texas Health Science Center’s Committee for Protection of Human Subjects (HSC-SPH-18-0233).

### 2.2. Measures

#### 2.2.1. Primary Outcome

Depressive symptoms were assessed by the Patient Health Questionnaire-2 (PHQ-2), a validated survey, in both English and Spanish [21,22,23]. PHQ-2 consists of two items screening the frequency of “having little interest or pleasure in doing things” or “feeling down, depressed, or hopeless” in the past two weeks on a Likert scale answer ranging from 0 to 3. A positive screening of depressive symptoms was indicated if the cutoff point ≥3 for combined PHQ-2 scores (ranging from 0 to 6). A PHQ-2 score ≥3 was identified as the optimal cutoff point with an 83% sensitivity and 92% specificity for major depression screening [21]. An internal consistency with a Cronbach’s alpha = 0.83 indicated good reliability of PHQ-2 [22].

Respondents’ depressive symptom occurrence status (Yes/No) was determined at the time of completion of the first and the second survey. Participants were categorized into two groups according to their depressive symptoms’ occurrence status at both surveys: (1) stable/improved (no depressive symptoms at both surveys, or depressive symptoms at the first survey and no depressive symptoms at the second survey), (2) consistent depressive symptoms/declined (no depressive symptoms at the first survey and depressive symptoms at the second survey, or depressive symptoms at both surveys).

#### 2.2.2. Sociodemographic Variables

Sociodemographic questions were self-reported and included age (≤40 years, >40 years), sex (female, male), race/ethnicity (non-Hispanic White, racial/ethnic minorities), education (less than high school/high school, college 1–3 years, college 4 years or more), household income in 2019 (under $25,000, $25,001–45,000, $45,001–65,000, $65,001 or greater), employment and wages (“no change due to COVID-19”: still employed and wages have stayed the same/retired or homemaker before and still; “temporary decrease in wages during COVID-19”: still employed and wages decreased during COVID-19 but now have increased again/lost job during COVID-19 but now employed again; “decrease in wages due to COVID-19”: still employed but wages have decreased/lost job during COVID-19 and still unemployed), the number of children in the household (none, one, two or more), and food security during COVID-19 (food secure, experienced food insecurity).

#### 2.2.3. Movement Behaviors

The survey included questions about the participant’s physical activity and sedentary time at various points during the COVID-19 pandemic. The participants’ change in physical activity level was reported as: maintained the same, increased, or decreased, compared to the first 3 months of the COVID-19 pandemic. In terms of sedentary time, participants reported how many hours/day for work and entertainment they were sedentary (0–2 h, 3–4 h, 5–6 h, or 7 or more hours). Participants were also asked to compare how this total sedentary time stayed the same, had increased, or was less than before the pandemic.

#### 2.2.4. Health Outcomes

The survey also assessed participants’ medical history. Specifically, there were questions about previously diagnosed chronic conditions, including diabetes, hypertension, heart disease, autoimmune conditions, or having undergone cancer treatments. A binary variable was generated to represent previously diagnosed chronic conditions if participants self-reported that they had at least one condition listed above. Additionally, there were questions on having been previously diagnosed with lung-related diseases/conditions, including chronic lung disease or moderate to severe asthma. A binary variable was created to indicate if they had at least one lung-related disease/condition.

### 2.3. Statistical Analysis

Descriptive statistics, including means, standard deviations, counts and percentages, were used to describe the occurrence of depressive symptoms, sociodemographic, behavioral, and health outcome characteristics among the sample. Chi-square tests (χ2) tests were performed to compare differences of categorical variables, and Kruskal-Wallis tests were used for not normally distributed continuous variables, respectively, across the depressive symptom occurrence patterns. Univariate logistic regression was performed between each variable and depressive symptom occurrence patterns, and those with *p* < 0.05 were entered into a final, multivariable logistic regression model. Odds ratios and 95% confidence intervals (95% CI) were estimated for the association between each independent variable and the dependent variable (depressive symptom occurrence patterns). All analyses were conducted using Stata (17.0, StataCorp LP, College Station, TX, USA).

## 3. Results

### 3.1. Sample Characteristics

Socio-demographic characteristics, movement behaviors, and chronic disease prevalence by depressive symptom occurrence patterns during the COVID-19 pandemic are depicted in Table 1. The analytic sample consisted of 290 individuals, of whom 13.5% were categorized as having consistent depressive symptoms/declined status. The sample was predominantly female (84.1%), and about half of the participants were over 40 years old (51.7%). Nearly three quarters of the sample identified as racial/ethnic minorities (71.0%). About one third of the sample had less than high school/high school educational level (33.8%), and 50% of the participants reported a household income in 2019 of $45,000 or less. About one-third of respondents had experienced decreased wages during the pandemic (32.8%). Regarding the number of children in the household, 22.3% and 40.1% of the respondents reported having one, and two or more child(ren) in their household, respectively. Moreover, half of the study participants had experienced food insecurity during the COVID-19 pandemic. Additionally, most of the respondents self-reported not having any diagnosed chronic diseases (79.5%), or previously diagnosed lung diseases (93.0%). There were statistically significant differences in depressive symptom occurrence patterns by race/ethnicity (*p* = 0.03) and age (*p* = 0.01).

### 3.2. Depressive Symptom Occurrence Patterns during COVID-19 and Socio-Demographic, Behavioral, and Health Outcome Factors

The results of univariate logistic regression analyses are presented in Table 2. Compared to non-Hispanic White participants, racial/ethnic minorities had 53% lower odds of having consistent depressive symptoms/declined status during the COVID-19 pandemic (OR: 0.47, 95% CI: 0.24–0.94). Individuals greater than 40 years old had 58% lower odds of having consistent depressive symptoms/declined status (OR: 0.42, 95% CI: 0.20–0.85), compared to those at aged 40 or younger. The odds of showing consistent depressive symptoms/declined status were 71% lower for those who had increased their physical activity level (OR: 0.29, 95% CI: 0.09–0.95), as opposed to those who maintained their physical activity level. On the other hand, those who reported spending 5–6 h/day in sedentary activities had higher odds of having consistent depressive symptoms/declined status, compared to those spending 0–2 h/day sedentary (OR: 3.55, 95% CI: 1.11–11.39).

Race/ethnicity, age, physical activity compared to the first 3 months of COVID-19, and sedentary time were included in the multivariable logistic regression model, which largely confirmed the results of the unadjusted models (Table 2). Collinearity diagnostics with a variance reflation factor (VIF) of 1.04 suggested that no multicollinearity was detected. Racial/ethnic minorities had 55% lower odds (AOR: 0.45, 95% CI: 0.21–0.94) of showing consistent depressive symptoms/declined status, compared to non-Hispanic Whites. Those older than 40 years old also had significantly lower odds (AOR: 0.26, 95% CI: 0.17–0.76) of consistent depressive symptoms/declined status, relative to those who were 40 years or younger. Furthermore, participants who increased their physical activity level during COVID-19, relative to those who maintained physical activity, had 73% lower odds of having consistent depressive symptoms/declined status (AOR: 0.27, 95% CI: 0.08–0.92). On the contrary, those who spent 5–6 h/day in sedentary activities had significantly higher odds of having consistent depressive symptoms/declined status, relative to those spending 0–2 h/day sedentary time (AOR: 3.91, 95% CI: 1.15–13.26). The Hosmer and Lemeshow goodness-of-fit test indicated that the logistic regression model fit the data adequately (*p* = 0.33).

## 4. Discussion

### 4.1. Summary of Findings

This study examined changes in the occurrence of depressive symptoms among a racially and ethnically diverse sample of adults during the COVID-19 pandemic, and explored the association between socio-demographic factors, movement behaviors, health history, and health behaviors with the occurrence of depressive symptoms patterns. Overall, between June and November 2020, the majority of the study sample were classified as having stable/improved depressive symptom occurrence status. However, there were several factors associated with depressive symptoms’ occurrence patterns. In particular, the results from the multivariable logistic regression model highlight the relevance of socio-demographic characteristics and movement behaviors for understanding depressive symptom occurrence patterns among the studied sample.

### 4.2. Socio-Demographic Factors

Race/ethnicity was a significant determinant of depressive symptoms’ status, and non-Hispanic White participants were more likely to have consistent depressive symptoms/declined status relative to racial/ethnic minorities during the COVID-19 pandemic. Our findings are inconsistent with the previous literature which typically shows that Hispanic individuals were more likely to be depressed compared to the non-Hispanic population at the start of COVID-19 [6,9]. A previous study also suggested that Hispanic and Asian vs. their counterparts reported higher levels of COVID-19 fear, which was associated with higher reported depression [5]. One longitudinal study among the US population suggested that no association between race/ethnicity and depression in multiple regression controlling for other socio-demographic variables [19]. Nevertheless, our study collected data in June and November 2020, while the previously mentioned longitudinal study was conducted between April and May 2020, which represented the early months after the emergence of COVID-19 before the dramatic upturn in COVID-19 cases, a relatively short-term pattern.

Moreover, previous studies show that depression was inversely related to assets, such as income, home ownership, marital status, and education, which are unequally distributed across all race/ethnicity groups [24]. Within their studied sample, minorities have fewer assets than non-Hispanic White individuals, and therefore, when adjusted for assets, Hispanics and non-Hispanic Black had a lower likelihood of probable depression relative to non-Hispanic White persons [24]. Interestingly, a recent study also found that, relative to non-Hispanic Whites, racial/ethnic minorities (Hispanics, non-Hispanic Blacks, and Asians) were less likely to report experiencing depression, even if they also tend to report having poorer health [25]. These differences could be explained by greater mental resilience [26], or by the Hispanic health paradox [27].

In this study, age was also significantly related to depressive symptoms’ status. Our findings are in line with the previous literature with longitudinal design suggests that older age was associated with a decreased likelihood of depression [14,19,28], and younger age is a predictor of depression in many cross-sectional studies among the US population [3,8,9,29] and other countries [30] during the COVID-19 pandemic. In addition, a study found that younger age is a persistent risk factor for depression during the pandemic [31]. One explanation could be that younger adults had a greater increase in unemployment and a greater decrease in working full-time relative to older age groups, compared to pre-pandemic times [32].

### 4.3. Movement Behaviors

A significant association was found between changes in physical activity, sedentary behavior, and depressive symptom occurrence patterns. Increased physical activity level was associated with a lower likelihood of showing consistent depressive symptoms/declined status. Furthermore, sedentary time of 5–6 h/day (vs. 0–2 h/day) significantly increased the likelihood of showing consistent depressive symptoms/declined status. Our finding was in accord with the literature, mostly in the form of cross-sectional studies, indicating the importance of physical activity in protecting against adverse mental health symptoms during the pandemic [17,33]. The direct association between sedentary behavior and depression symptoms has also been reported in the literature before [34] and during the pandemic [17,32].

Our findings are broadly in line with a recent longitudinal study in Germany conducted by Mata et al. suggesting that depression was inversely related to physical activity and directly associated with screen time (hours/day) [17]. Physical activity brings benefit against depression through physiological changes, e.g., anti-inflammatory and oxidative stress, and psychological mechanisms, such as enhanced self-esteem and social support [35]. It is noteworthy that the aforementioned German study used the mean score of two items of PHQ-2 as the dependent variable for depression, while the current study calculated the total score of the two-items and classified depression based on the optimal cutoff point of 3, which was validated with a sensitivity of 83% and specificity of 92% for screening of major depression [21].

### 4.4. Strengths and Limitations of Study

There are several strengths in the current study. First, the longitudinal study design allowed us to capture the changes in depressive symptoms using a validated instrument between the early months and 6 months later in 2020 during the pandemic. Since the majority of research has utilized a cross-sectional design, the current study makes a valuable contribution to the literature. We also utilized a diverse sample that represents race/ethnicity minorities. Additionally, the study considered a variety of factors that could potentially affect individuals’ depressive symptoms’ status.

Nevertheless, some limitations should be considered. First, participants were categorized into only two groups because of the limited sample size, which hindered our ability to explore whether factors associated with depression onset differed from those required for maintenance of depressive symptoms. Second, our sample was recruited in Central Texas. Even though COVID-19 is a global pandemic, the state and local levels of severity and policy differences in our two-measurement time points might limit the generalizability to other geographic areas. Similarly, the study data were only collected in June and November 2020, respectively; therefore, the relationship between the determinants and depressive symptoms’ status may alter regarding changes in the pandemic in Central Texas. Further, the sample was predominantly female (84.1%), which may not generalize to men. Besides, all the information was self-reported by participants, which may be affected by recall bias and social desirability bias. Lastly, PHQ-2 is a validated instrument for depression screening, and it is a desirable measure that limits participants’ survey burden. However, it does not reflect the severity of depression. Other more robust measures, such as PHQ-9, that could potentially provide more accurate and detailed information should be considered in future studies.

## 5. Conclusions

In conclusion, the current longitudinal study identified several determinants of change in depressive symptoms’ status between the first 3 months and 8 months of the COVID-19 pandemic in Central Texas among a racially/ethnically diverse sample. Race/ethnicity is a crucial factor that affects the depressive symptoms status between two-time points. Nevertheless, in contrast to the association documented in the previous literature, the current study found that racial/ethnic minorities were less likely to have consistent depressive symptoms/declined status relative to non-Hispanic White individuals. Further explorations are needed to examine the relation between race/ethnicity and depressive symptoms, especially within the context of the COVID-19 pandemic and keeping in mind structural inequalities around race and ethnicity in the US. The pandemic appears to have had a greater effect on depression among younger adults. Providing resources and guidance for younger adults to cope with depression is urgently needed. Behavioral factors also play important roles in affecting depression during the COVID-19 pandemic. Engaging in greater physical activity could be one of the strategies that protect against adverse mental health, while longer total sedentary hours were associated with negative mental health outcomes.

After 20 months since COVID-19’s arrival on Central Texas, we are still co-existing with the pandemic situation. As we continue to traverse this new normal, the subsequent and long-term effects of the pandemic on physical and mental health in all walks of life remain unclear, which should be examined in future studies. Measuring mental health at multiple time points among a large and racially/ethnically diverse sample will be beneficial to understand the trends in depression patterns and identify the key determinants for different populations.

## Figures and Tables

**Table 1 ijerph-19-01194-t001:** Demographic, behavioral and health outcome variables during COVID-19 among central Texas residents by depressive symptoms’ occurrence patterns, FRESH-Austin Study, 2020 (*n* = 290).

Variable	Total	Stable/Improved ^a^	Consistent Depressive Symptoms/Declined ^b^	*p*-Value
	*n* (%)	*n* (%)	*n* (%)	
Total	290 (100)	251(86.6)	39 (13.5)	
Race/Ethnicity				
non-Hispanic White	84 (29.0)	67 (26.7)	17 (43.6)	0.03 *
Racial/Ethnic Minorities	206 (71.0)	184 (73.3)	22 (56.4)	
Age (year)				
≤40	140 (48.3)	114 (45.4)	26 (66.7)	0.01 *
>40	150 (51.7)	137 (54.6)	13 (33.3)	
Sex				
Female	243 (84.1)	210 (83.7)	33 (86.8)	0.62
Male	46 (15.9)	41 (16.3)	5 (13.2)	
Education				
Less than high school/high school	98 (33.8)	87 (34.7)	11 (28.2)	0.72
College 1–3 years	57 (19.7)	49 (19.5)	8 (20.5)	
College 4 years or more	135 (46.6)	115 (45.8)	20 (51.3)	
Income 2019				
Under 25k	66 (22.8)	57 (22.7)	9 (23.1)	0.98
$25,001–45,000	79 (27.2)	69 (27.5)	10 (25.6)	
$45,001–65,000	46 (15.9)	39 (15.5)	7 (18.0)	
$65,001 or greater	99 (34.1)	86 (34.3)	13 (33.3)	
Employment status and wages				
No change due to COVID-19 ^c^	149 (51.6)	133 (53.2)	16 (41.0)	0.26
Temporary decrease in wages during COVID-19 ^d^	45 (15.6)	36 (14.4)	9 (23.1)	
Decrease in wages due to COVID-19 ^e^	95 (32.8)	81 (32.4)	14 (35.9)	
Child(ren) in household				
None	108 (37.6)	89 (35.9)	19 (48.7)	0.27
One	64 (22.3)	58 (23.4)	6 (15.4)	
Two or more	115 (40.1)	101 (40.7)	14 (35.9)	
Food Secure				
Food Secure during COVID-19	145 (50.0)	129 (51.4)	16 (41.0)	0.23
Experienced food insecurity during COVID-19	145 (50.0)	122 (48.6)	23 (59.0)	
Physical activity compared to the first 3 months of COVID-19				
Same	79 (27.2)	66 (26.3)	13 (33.3)	0.05
Decrease	138 (47.6)	116 (46.2)	22 (56.4)	
Increase	73 (25.2)	69 (27.5)	4 (10.3)	
Sedentary behavior change				
About the same now as before COVID	159 (55.4)	134 (53.6)	25 (67.6)	0.29
Sitting less before COVID	86 (30.0)	77 (30.8)	9 (24.3)	
Sitting more before COVID	42 (14.6)	39 (15.6)	3 (8.1)	
Sedentary time (hours)				
0–2 h	74 (25.5)	69 (27.5)	5 (12.8)	0.16
3–4 h	64 (22.1)	56 (22.3)	8 (20.5)	
5–6 h	44 (15.2)	35 (13.9)	9 (23.1)	
7 or more hours	108 (37.2)	91 (36.3)	17 (43.6)	
Previous diagnosed chronic conditions				
No	221 (79.5)	191 (79.6)	30 (78.9)	0.93
Yes	57 (20.5)	49 (20.4)	8 (21.1)	
Previous lung diseases/conditions				
No	267 (93.0)	229 (92.3)	38 (97.4)	0.21
Yes	20 (7.0)	19 (7.7)	1 (2.6)	
PHQ-2 Score (mean, SD)				
June 2020 (first survey)	1.35 (1.6)	1.09 (1.3)	3.03 (2.0)	0.00 **
November 2020 (second survey)	1.32 (1.5)	0.84 (0.9)	4.36 (1.3)	0.00 **

Note: ^a^ stable/improved: individuals who had no depressive symptoms at both surveys, or depressive symptoms at the first survey and no depressive symptoms at the second survey; ^b^ consistent depressive symptoms/declined: individuals who had no depressive symptoms at the first survey and depressive symptoms at the second survey, or depressive symptoms at both surveys; ^c^ still employed and wages have stayed the same/retired or homemaker before and still; ^d^ still employed and wages decreased during COVID-19 but now have increased again/lost job during COVID-19 but now employed again; ^e^ still employed but wages have decreased/lost job during COVID-19 and still unemployed. Missing value: Gender (*n* = 1), Employment status and wages (*n* = 1), Child(ren) in household (*n* = 3), Previous diagnosed diseases (*n* = 12), Previous lung diseases (*n* = 3), Sedentary behavior change (*n* = 3). SD: Standard Deviation, *p* < 0.05 *, *p* < 0.01 **.

**Table 2 ijerph-19-01194-t002:** Univariate and multivariable logistic regression of the correlates of depressive symptom occurrence patterns during COVID-19 pandemic among central Texas residents, FRESH-Austin Study, 2020 (*n* = 290).

Referent = Stable/Improved ^a^	Consistent Depressive Symptoms/Declined ^b^			
Variable	Unadjusted OR (95% CI)	*p*-Value	Adjusted OR (95% CI)	*p*-Value
Race/Ethnicity				
non-Hispanic White (referent)	1.0		1.0	
Racial/Ethnic Minorities	0.47 (0.24–0.94)	0.03 *	0.45 (0.21–0.94)	0.04 *
Age (year)				
≤40 (referent)	1.0		1.0	
>40	0.42 (0.20–0.85)	0.02 *	0.26 (0.17–0.76)	0.01 **
Sex				
Female (referent)	1.0			
Male	0.78 (0.29–2.11)	0.62		
Education				
Less than high school/high school (referent)	1.0			
College 1–3 years	1.29 (0.49–3.43)	0.61		
College 4 years or more	1.38 (0.63–3.02)	0.43		
Income 2019				
Under 25k (referent)	1.0			
$25,001–45,000	0.92 (0.35–2.41)	0.86		
$45,001–65,000	1.14 (0.39–3.31)	0.81		
$65,001 or greater	0.96 (0.38–2.39)	0.93		
Employment status and wages				
No change due to COVID-19 (referent) ^c^	1.0			
Temporary decrease in wages during COVID-19 ^d^	2.08 (0.85–5.09)	0.11		
Decrease in wages due to COVID-19 ^e^	1.44 (0.67–3.10)	0.36		
Child(ren) in household				
None (referent)	1.0			
One	0.48 (0.18–1.29)	0.15		
Two or more	0.65 (0.31–1.37)	0.26		
Food security				
Food Secure during COVID-19 (referent)	1.0			
Experienced food insecurity during COVID-19	1.52 (0.77–3.01)	0.23		
Physical activity compared to the first 3 months of COVID-19				
Same (referent)	1.0		1.0	
Decrease	0.96 (0.46–2.04)	0.92	0.83 (0.38–1.82)	0.64
Increase	0.29 (0.09–0.95)	0.04 *	0.27 (0.08–0.92)	0.04 *
Sedentary behavior change				
About the same now as before COVID (referent)	1.0			
Sitting less before COVID	1.6 (0.71–3.59)	0.26		
Sitting more before COVID	0.66 (0.17–2.57)	0.55		
Sedentary time (hours)				
0–2 h (referent)	1.0		1.0	
3–4 h	1.97 (0.61–6.36)	0.26	1.45 (0.43–4.9)	0.55
5–6 h	3.55 (1.11–11.39)	0.03 *	3.91 (1.15–13.26)	0.03 *
7 or more hours	2.58 (0.91–7.33)	0.08	1.75 (0.58–5.28)	0.32
Previous diagnosed chronic conditions				
No (referent)	1.0			
Yes	1.04 (0.45–2.41)	0.93		
Previous lung diseases/conditions				
No (referent)	1.0			
Yes	0.32 (0.04–2.44)	0.27		

Note: ^a^ stable/improved: individuals who had no depressive symptoms at both surveys, or depressive symptoms at the first survey and no depressive symptoms at the second survey; ^b^ consistent depressive symptoms/declined: individuals who had no depressive symptoms at the first survey and depressive symptoms at the second survey, or depressive symptoms at both surveys; ^c^ still employed and wages have stayed the same/retired or homemaker before and still; ^d^ still employed and wages decreased during COVID-19 but now have increased again/lost job during COVID-19 but now employed again; ^e^ still employed but wages have decreased/lost job during COVID-19 and still unemployed. SD: Standard Deviation, *p* < 0.05 *, *p* < 0.01 **.

## Data Availability

Data can be shared by request upon contacting the authors.

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
