# Peer review of "Change in Depression and Its Determinants during the COVID-19 Pandemic: A Longitudinal Examination among Racially/Ethnically Diverse US Adults"

_ijerph, 2022, doi:10.3390/ijerph19031194_

Round 1

Reviewer 1 Report

The study examined longitudinal data to identify changes in the occurrence of depressive symptoms, and to explore if such changes were associated with socio-demographic, movement behaviors, and health variables during the COVID-19, among a diverse sample of central Texas residents. The topic of the manuscript is within the scope of the Journal and could be valuable to the scientific audience. The quality of the research design is acceptable.

TITLE

The title of the article is accurate.

ABSTRACT

Abstract reflects the work done and the conclusions drawn.

INTRODUCTION

Authors should formulate research hypothesis (-es).

 METHOD

Some clarifications are however needed.

Why is it not justified whether assumptions for using ANOVA were met (normality, multicollinearity, homogeneity of variance)?

RESULTS

The results section is missing some information. Have some other assumptions for logistic regressions analysis been tested and met (For instance, multicollinearity (no high intercorrelations among the predictors).

DISCUSSION

Please discuss the confirmation or rejection of research hypothesis(-es).

Predominantly female sample (84.1%) state as a limitation.

 TO SUM UP I think the author(s) need to make the recommended corrections.

Reviewer 2 Report

I am not a content expert. As a non-expert reader, I tried to evaluate whether the manuscript can be read easily. In my opinion, this manuscript can be easily read and can be understood. This study conducted a longitudinal study to examine correlates of changes in depressive symptoms over 5 months in a sample of participants with a diverse ethical backgrounds. What is new in the study and why knowing the current results is important are clearly written. The methodology is sound. Results are sufficiently written. Tables are helpful. Discussion is complete. 

However, I believe that another limitation should be described. Categorizing the participants into only two groups may be an important limitation. This is because doing so will make it difficult to identify factors which cause depressive symptoms and maintain these symptoms. 

Reviewer 3 Report

I think that this is a well written article that appears to have been conducted to an adequate standard warranting publication. I think that the findings will be of interest to researchers in this field and have only a small number of questions for the authors:

My chief issue with the manuscript is that the study was conducted using only a 2-item questionnaire to measure depressive symptoms. This leaves a lot of uncertainty and impreciseness as to the true depressive symptoms of the cohort and can make it hard to differentiate from other difficulties experienced during this time. I know that there is nothing that can be done about this factor at this point – and I think that the study still has merit – but I think that it would be worth (a) describing the validity/reliability metrics associated with the PHQ-2 in the methods and (b) noting that this is a limitation in the discussion.

Were the tests corrected for multiple comparisons? There were a lot of tests run and this would be appropriate given the number of variables

Is there any indication whether the consistent/decline in depressive symptoms is any different to what would happen outside of COVID pandemic? Could the authors please demonstrate that this differs from what a similar population would report on this outcome measure outside of COVID times, if possible to put the effect of COVID on depression overall into some context?

Could inter-state differences in culture/socio-economic environment contribute to the differences between the findings in the current study and [19] (as well as other similar studies), perhaps especially in terms of the ethnicity differences?

Would it be beneficial to attempt to covary for assets/financial situation in the regressions in the current data, as in [24]?
